# Learning Instruction-Following Policies through Open-Ended Instruction Relabeling with Large Language Models

## Abstract

Developing effective instruction-following policies in reinforcement learning remains challenging due to the reliance on extensive human-labeled instruction datasets and the difficulty of learning from sparse rewards. In this paper, we propose a novel approach, **O**pen-ended **I**nstruction **R**elabeling (**OIR**), that leverages the capabilities of large language models (LLMs) to automatically generate open-ended instructions retrospectively from previously collected agent trajectories. Our core idea is to employ LLMs to relabel unsuccessful trajectories by identifying meaningful subtasks the agent has implicitly accomplished, thereby enriching the agent's training data and substantially alleviating reliance on human annotations. Through this open-ended instruction relabeling, we efficiently learn a unified instruction-following policy capable of handling diverse tasks within a single policy. We empirically evaluate our proposed method in the challenging Craftax environments, demonstrating clear improvements in sample efficiency, instruction coverage, and overall policy performance compared to state-of-the-art baselines. Our results highlight the effectiveness of utilizing LLM-guided open-ended instruction relabeling to enhance the instruction-following abilities through reinforcement learning. The code is available at https://anonymous.4open.science/r/ICLR26-OIR/.

## 1 Introduction

Instruction-following reinforcement learning (RL), where agents learn to efficiently interpret and execute tasks specified through natural-language instructions, holds immense promise for building generalizable and flexible AI systems. Despite considerable advances in goal-conditioned RL methods (Schaul et al., 2015; Andrychowicz et al., 2017), existing instruction-following RL approaches continue to face significant challenges. Typically, such methods heavily rely on large-scale human-annotated instruction datasets (Hill et al., 2020; Narasimhan et al., 2018) or predefined instruction templates, limiting scalability, generalization capabilities, and thereby constraining their real-world applicability. Furthermore, environments characterized by sparse feedback exacerbate this challenge—agents are likely to collect numerous unsuccessful trajectories that offer minimal utility, leading to inefficient exploration and slow policy improvement.

In this paper, we propose a novel framework to address these challenges by leveraging the strong reasoning capabilities of pretrained large language models (LLMs). Our key insight is to apply LLMs retrospectively to generate meaningful, open-ended instructions from collected agent trajectories. Specifically, the LLM identifies semantically relevant subtasks that the agent implicitly accomplished within failed trajectories and provides corresponding instructions, thus enriching these experiences with informative reward signals. Through this automatic instruction relabeling, we efficiently transform previously sparse and unsuccessful trajectories into valuable learning samples, thereby improving the data efficiency and diversity of instruction-conditioned policy learning without requiring any manual annotation effort.

We empirically validate our method on Craftax (Matthews et al., 2024), a challenging benchmark that offers diverse semantic instructions and inherently sparse rewards. Experimental results demonstrate that our proposed framework significantly surpasses strong baselines, achieving substantial improve-

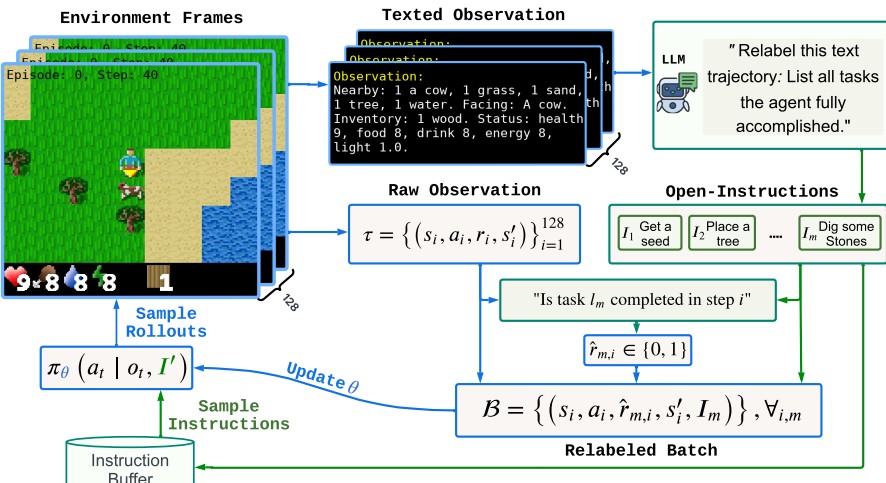

Figure 1: Overview of **O**pen-ended **I**nstruction **R**elabeling (**OIR**) framework. → **Blue flow** → illustrates standard reinforcement learning: an instruction-conditioned policy samples rollouts from the environment using instructions sampled from the instruction buffer, after which the policy parameters are updated with the collected samples. → **Green flow** → highlights our novel relabeling mechanism: converting collected trajectories into extual observations and then prompting a pretrained LLM to retrospectively generate diverse, open-ended instructions identifying successfully accomplished subtasks. These generated instructions then serve to relabel trajectories by producing binary, semantic rewards, thereby enriching the instruction buffer with new and informative learning signals.

ments in terms of sample efficiency, coverage and diversity of instructions, and the overall quality of learned instruction-following policies. Our primary contributions can be summarized as follows:

1. We propose a novel open-ended instruction relabeling framework that leverages large language models to automatically assign semantically meaningful instructions to collected trajectories, completely eliminating dependence on manual human annotations.

2. Our relabeling method effectively transforms unsuccessful trajectories into informative training examples, enabling more efficient learning of instruction-following policies in open-ended environments.

3. Extensive empirical experiments conducted on the Craftax benchmark validate that our approach substantially outperforms state-of-the-art instruction-conditioned reinforcement learning baselines, demonstrating strong improvements in sample efficiency, instruction diversity, and instruction-following performance.

## 2 RELATED WORK

**Goal-Conditioned Reinforcement Learning (GCRL).** Goal-conditioned reinforcement learning generalizes standard RL objectives by conditioning policies and reward functions explicitly on goal representations (Schaul et al., 2015; Liu et al., 2022). Many GCRL methods leverage hindsight relabeling strategies to address sparse rewards, notably Hindsight Experience Replay (HER) (Andrychowicz et al., 2017). HER-inspired methods propose various heuristics to select suitable goals, such as goal discovery based on reward relevance (Pitis et al., 2020; Fang et al., 2019), goal diversity (Ren et al., 2019), and adaptive difficulty selection based on learning progress (Nair et al., 2018; Warde-Farley et al., 2018). However, existing methods primarily rely on predefined numeric or state-based goals and cannot easily scale to instructions presented in open-ended natural language.

**Instruction-Conditioned Reinforcement Learning.** Instruction-conditioned RL extends goal-conditioned RL by formulating tasks explicitly via natural language instructions that are encoded directly into policy inputs and rewards (Luketina et al., 2019; Narasimhan et al., 2018; Hill et al., 2020). Recent works propose using hindsight instruction relabeling (HIR) (Zhang et al., 2023; Sumers et al., 2023; Xiao et al., 2022) methods that retrospectively assign appropriate instruction labels to collected

trajectories, substantially enhancing data efficiency and policy robustness. Nonetheless, these HIR approaches typically require human-defined templates (Sumers et al., 2023) or substantial manual labeling efforts (Zhang et al., 2023; Xiao et al., 2022), limiting their scalability and generalization potential. Our approach, in contrast, eliminates the need for manually defined instruction spaces or labels by using LLM-generated relabeling, significantly increasing flexibility and open-ended task coverage.

**Leveraging Large Language Models in Reinforcement Learning.** Recent work has increasingly explored the use of pretrained Large Language Models (LLMs) to assist various aspects of RL tasks, including semantic reward shaping (Xie et al., 2023; Ma et al., 2023), task decomposition (Huang et al., 2022; Lin et al., 2022), and high-level action guidance (Du et al., 2023; Yao et al., 2023; Fan et al., 2022; Wang et al., 2025). Such methods exploit the reasoning power and semantic understanding capabilities of LLMs to enrich RL policy learning, although they often focus on forward generation of guidance or rewards from human-written task descriptions. By contrast, our proposed approach uniquely applies LLMs in a retrospective fashion: generating meaningful instruction labels from collected trajectory interactions, thus enabling effective utilization of unsuccessful episodes and improving sample efficiency and generalization across open-ended natural language instructions.

## 3 PRELIMINARIES

**Markov Decision Process (MDP).** A Markov Decision Process (MDP) provides a mathematical framework for modeling sequential decision-making problems (Puterman, 1994; Sutton & Barto, 2018). Formally, an MDP is defined as a tuple $(\mathcal{S}, \mathcal{A}, T, R, \gamma)$, where $\mathcal{S}$ is the state space, $\mathcal{A}$ is the action space, $T : \mathcal{S} \times \mathcal{A} \to \mathcal{S}$ describes the state transition dynamics, $R : \mathcal{S} \times \mathcal{A} \to \mathbb{R}$ is the reward function, and $\gamma \in [0, 1)$ is the discount factor. The goal of an agent interacting with the MDP is to find a policy $\pi : \mathcal{S} \to \mathcal{A}$ that maximizes the expected cumulative discounted reward:

$$J(\pi) = \mathbb{E}_{s_0 \sim p(s_0), a_t \sim \pi(s_t)} \left[ \sum_{t=0}^{\infty} \gamma^t R(s_t, a_t) \right]. \tag{1}$$

**Hindsight Experience Replay (HER).** Reinforcement learning algorithms often struggle in sparse reward environments, as successful experiences occur infrequently. Hindsight Experience Replay (HER) (Andrychowicz et al., 2017) addresses this challenge by enabling agents to learn effectively from failures through goal relabeling. Specifically, HER retrospectively reinterprets unsuccessful episodes by setting the goals to states the agent actually achieved later in the trajectory.

Formally, given an observed trajectory $\tau = \{(s_t, a_t, r_t, s_{t+1}, g)\}_{t=0}^{T-1}$ associated with an original goal $g$, HER selects an achieved future state $s_{t'}$ (where $t' > t$) from the trajectory and relabels the original goal as $g' = s_{t'}$. The reward for the relabeled trajectory is then recomputed accordingly:

$$r'_t = R(s_t, a_t, g'). \tag{2}$$

By using achieved states as hindsight goals, HER significantly enhances the agent's ability to learn efficiently from sparse rewards, effectively converting unsuccessful episodes into valuable learning experiences.

## 4 METHOD

We propose a novel approach leveraging hindsight instruction relabeling guided by large language models (LLMs) to efficiently train generalizable instruction-conditioned reinforcement learning (RL) policies. Unlike traditional hindsight experience replay techniques (Andrychowicz et al., 2017), which reuse visited states to generate goals, our method synthesizes diverse free-form textual instructions directly from collected trajectories. This synthesis allows agents to learn from a richer set of instructions without requiring domain-specific knowledge, thereby facilitating effective training in environments characterized by sparse and semantic reward signals.

We formalize instruction-following RL tasks as an MDP extended with an explicit instruction space $(\mathcal{S}, \mathcal{A}, \mathcal{I}, R, T, \gamma)$, where $\mathcal{S}$ is the state space, $\mathcal{A}$ the action space, and $\mathcal{I}$ is the (potentially unbounded) textual instruction space expressed in natural language.

---

**Algorithm 1 OIR**: LLM-Guided Hindsight Instruction Relabeling

---

**Require:** pretrained LLM $\mathcal{L}$, encoders $f_{\text{state}}$, $f_{\text{instr}}$, off-policy RL algorithm `alg`
1: Initialise instruction buffer $\mathcal{B}$ and $E$ parallel environments
2: **for** each iteration **do**
3:     $\mathcal{D} \leftarrow \emptyset$
4:     Collect trajectories $\{\tau_e\}_{e=1}^E$ with policy $\pi_\theta$
5:     **for** $e = 1$ **to** $E$ **do**
6:         $\texttt{prompt}_e \leftarrow h(\tau_e)$                                       ▷ build LLM prompt
7:         **for** $k = 1$ **to** $K$ **do**
8:             $i'_{e,k} \sim \mathcal{L}(\texttt{prompt}_e)$                          ▷ generate candidate
9:             $r_{e,k}^{\text{cand}} \leftarrow R\big(i'_{e,k}, \tau_e\big)$            ▷ cosine-similarity reward, Eq. 7
10:            $\mathcal{D} \leftarrow \mathcal{D} \cup \{(\tau_e, i'_{e,k}, r_{e,k}^{\text{cand}})\}$
11:        **Update** $\mathcal{B}$ with all the relabeled instructions $\{i'_{e,k} \mid e \in [E], k \in [K]\}$ using Section 4.3
12:        Update $(\theta, \psi)$ using batch $\mathcal{D}$ with `alg`
13:        Reset finished environments with instructions sampled uniformly from $\mathcal{B}$

---

At each timestep $t$, the agent observes a state $s_t \in \mathcal{S}$ and an instruction $i \in \mathcal{I}$. It then selects an action $a_t \in \mathcal{A}$ according to a conditional policy $a_t \sim \pi_\theta(a_t \mid s_t, f_{\text{instr}}(i))$, where $f_{\text{instr}}(\cdot)$ is an embedding function encoding textual instructions into embedding vectors. In our setup, we assume access to a pretrained instruction embedding encoder $f_{\text{instr}}$, such as SBERT (Reimers & Gurevych, 2019).

Subsequently, the environment transitions to the next state $s_{t+1} \sim T(s_t, a_t)$ and produces a reward explicitly conditioned on the current instruction $r_t = R(s_t, a_t, i)$.

We assume that the ground truth reward function $R$ is binary (with $R(s_t, a_t, i) = 1$ indicating successful completion and $0$ otherwise), but typically not acessible for arbitrary instructions during training. To address this challenge, we use hindsight instruction relabeling with automatically generated instructions from LLMs, effectively synthesizing surrogate binary reward signals based on collected trajectories to enable efficient learning.

Finally, we express our overall training and evaluation objective through a statistical mapping $f$, aggregating individual instruction-conditioned returns (or success indicators) into a single scalar metric:

$$\max_\theta f\left(\mathbb{E}_{i \sim \mathcal{I}}\left[G(i, \pi_\theta)\right]\right), \tag{3}$$

where $G(i, \pi_\theta)$ denotes the instruction-conditioned cumulative reward or binary achievement indicator. Specific instantiations of the aggregation function $f$ include mean cumulative reward across instructions (expected return), success rate, and aggregated achievement score. We describe and evaluate these metrics concretely in Section 5.

Detailed in Algorithm 1, our method is composed of: (1) the instruction generation and relabeling procedure; (2) reward and episode termination assignment based on the new instructions; and (3) the prioritized instruction buffer used to efficiently manage the instructions utilized during rollouts.

### 4.1 TRAJECTORY COLLECTION AND INSTRUCTION RELABELING VIA LLMS

During training, we concurrently deploy $E$ parallel instances of the environment, each collecting trajectories under policy $\pi_\theta(\cdot \mid o_t, i_t)$. We denote the set of collected trajectories as $\{\tau_e\}_{e=1}^E$, where each trajectory is a sequence consisting of state-action pairs $\tau_e = \{(s_t^e, a_t^e)\}_{t=0}^{T_e}$.

To generate meaningful instructions from these trajectories, we leverage the capacity of large language models to perform semantic reasoning over textual descriptions of observed interactions. Concretely, we first convert each trajectory $\tau_e$ into an interpretable textual format suitable for prompting a pretrained LLM $\mathcal{L}$. Formally, given a trajectory $\tau_e$, we construct a textual prompt structured temporally as follows:

```
Prompt:
What instruction is this trajectory following?
timestep 0:  textual observation {{o_0^e}}, agent takes action {{a_0^e}}
timestep 1:  textual observation {{o_1^e}}, agent takes action {{a_1^e}}
...
timestep T_e:  textual observation {{o_{T_e}^e}}, agent takes action {{a_{T_e}^e}}
```

Using such temporally structured prompts, the LLM returns plausible instructions corresponding to the actions of the trajectory. In general, given the language model $\mathcal{L}$, a trajectory-based textual observation transformation function $h$, and hyperparameter $K$ controlling instruction quantity and diversity, the instruction generation process is given by:

$$\{i_{e,k}\}_{k=1}^K \sim \mathcal{L}(\texttt{prompt}(\tau_e)), \tag{4}$$

with `prompt` being the prompt template shown above.

Thus, each trajectory yields a set of $K$ candidate instructions suitable for the corresponding behavior. These synthetically generated instructions are subsequently used to generate synthetic reward and termination signals.

**Relabeling Failed Trajectories.** An important property of our method is its ability to explicitly leverage and reinterpret trajectories that fail under their originally assigned instructions. Even when the agent does not successfully achieve the intended task during data collection, the hindsight relabeling procedure can still extract meaningful learning signals by deriving alternative instructions associated with the same behavioral sequence.

We evaluate relabeled instructions by asking whether an oracle agent that can perfectly follow any instruction would score the observed behavior as good for that instruction. For any instruction $i$ with encoding $f_{\text{instr}}(i)$, discount $\gamma \in (0,1)$, transition kernel $T$, and initial state distribution $p(s_0)$, define:

$$\pi_i^{\text{oracle}} \in \arg\max_\pi \mathbb{E}\left[\sum_{t=0}^\infty \gamma^t R(s_t, a_t, i) \,\middle|\, s_0 \sim p,\ a_t \sim \pi(\cdot \mid s_t, f_{\text{instr}}(i)),\ s_{t+1} \sim T(\cdot \mid s_t, a_t)\right],$$

$$V_i^{\text{oracle}}(s) = \mathbb{E}\left[\sum_{t=0}^\infty \gamma^t R(s_t, a_t, i) \,\middle|\, s_0 = s,\ a_t \sim \pi_i^{\text{oracle}}(\cdot \mid s_t, f_{\text{instr}}(i)),\ s_{t+1} \sim T(\cdot \mid s_t, a_t)\right]. \tag{5}$$

Let the action value be $Q_i^{\text{oracle}}(s,a) := R(s,a,i) + \gamma \mathbb{E}_{s' \sim T(\cdot \mid s,a)}[V_i^{\text{oracle}}(s')]$. A candidate instruction $i'$ proposed at step $t$ for the observed pair $(s_t, a_t)$ is effective if

$$Q_{i'}^{\text{oracle}}(s_t, a_t) > Q_{i_{\text{orig}}}^{\text{oracle}}(s_t, a_t). \tag{6}$$

This stepwise test supports partial instruction following, since an instruction can be rewarded even if only part of the trajectory would be optimal under it. In practice, the oracle is unavailable; we use a language model to propose such effective candidates.

Language models may propose inaccurate or misleading instructions because they lack environment-specific context and can struggle with long trajectories. We therefore add simple rule-based instructions as a second supervision signal. This also motivates the subsequent design of reward function and instruction replay buffer, which are designed to ensure the quality of the relabeled instructions.

## 4.2 REWARD DEFINITION AND EPISODE TERMINATION CRITERION

To flexibly handle diverse, free-form instructions, we utilize an embedding-based reward function defined by semantic cosine similarity following ELLM (Gallici et al., 2024). Let $f_{\text{state}}(o_t)$ denote the embedding of transition $(o_t, a_t, o_{t+1})$ and $f_{\text{instr}}(i)$ the embedding of instruction $i$. At each time step $t$, the reward for instruction completion is given by

$$r_t(o_t, i) = \text{cosim}(f_{\text{state}}(o_t, a_t, o_{t+1}), f_{\text{instr}}(i)), \tag{7}$$

where the cosine similarity is defined as $\text{cosim}(\mathbf{a}, \mathbf{b}) = \frac{\mathbf{a} \cdot \mathbf{b}}{\|\mathbf{a}\| \|\mathbf{b}\|}$. An episode is deemed successful the first time the reward exceeds a predefined threshold $\delta$, $r_t(o_t, i) > \delta$, at which point the episode is marked as terminated.

## 4.3 PRIORITIZED INSTRUCTION REPLAY BUFFER

To handle the enlarged instruction set while keeping memory bounded, we adopt an eviction–rather than a sampling–based replay strategy inspired by Prioritized Level Replay (PLR) (Jiang et al., 2021). For every instruction $i$ we maintain its empirical mean return $\bar{R}(i)$, which are used only to *order* new instructions before they are inserted; afterwards, instructions are drawn uniformly at random from the buffer.

**Priority ordering.** We assign each instruction to one of three categories,

$$\text{Status}(i) = \begin{cases} 0 = \text{Learning boundary}, & \tau_{\text{low}} < \bar{R}(i) \leq \tau_{\text{high}}, \\ 1 = \text{Failing}, & \bar{R}(i) \leq \tau_{\text{low}}, \\ 2 = \text{Mastered}, & \bar{R}(i) > \tau_{\text{high}}. \end{cases} \tag{8}$$

where $0 < \tau_{\text{low}} < \tau_{\text{high}}$ are fixed thresholds. Learning boundary items are most valuable because they reveal the agent's frontier of competence. Within each category, we break ties by $\bar{R}(i)$, preferring instructions seen fewer times. Formally, we sort candidates by their status and empirical return:

$$\big(\text{Status}(i), \bar{R}(i)\big), \qquad \text{Status}(i) \in \{0, 1, 2\}, \tag{9}$$

where lexicographically smaller tuples have higher eviction priority.

**Round-robin eviction.** The replay buffer is a fixed-size circular array. After sorting, we iterate through the ordered list of new instructions and insert them sequentially, overwriting existing entries in round-robin fashion. In this way the buffer always contains the most recent instructions from the top of the priority list while still retaining a mixture of older tasks.

**Uniform sampling at episode termination.** Whenever an environment reaches the end of an episode, the next instruction is selected uniformly at random from the buffer. Because the buffer composition is itself prioritized, this simple sampling rule suffices to focus training on tasks close to the agent's learning boundary while still providing occasional exposure to harder and already mastered instructions.

This eviction-based replay mechanism balances exploration (by continually introducing under-sampled or failing tasks) and exploitation (by frequently retaining learning-boundary tasks), leading to faster and more robust policy improvement without the need for complicated probability computations or importance-sampling corrections.

## 5 EXPERIMENTS

We empirically evaluate our proposed method to assess its capability in learning instruction-conditioned RL policies, specifically addressing three central research questions.

- **RQ1 (Efficiency):** Does the integration of LLM-guided hindsight instruction relabeling improve training efficiency compared to baseline methods?
- **RQ2 (Generalization):** Can our method generalize beyond training instructions and successfully handle previously unseen instructions through semantic supervision obtained from LLM-generated instruction relabeling?
- **RQ3 (Diversity):** Does our approach increase the semantic diversity and coverage of instructions compared to baseline methods?

**Environment.** We adopt the Craftax-Classic environment (Matthews et al., 2024), an open-ended reinforcement learning benchmark that provides diverse, procedurally-generated environments explicitly characterized by textual instructions (*achievements*), which makes it particularly suitable for evaluating instruction-conditioned RL methods. Craftax-Classic originally offers sparse achievement notifications linked explicitly to environment-defined tasks (e.g., *"collect wood"*). However, we completely remove all built-in rewards and environment-provided achievement signals during training, making it necessary for the agent to rely exclusively on its own to explore the open-ended environment.

**Training Protocol.** We conduct all experiments using *parallelised Q-network* (PQN) (Gallici et al., 2024) as our base RL algorithm. PQN enables efficient sampling across multiple parallel environment

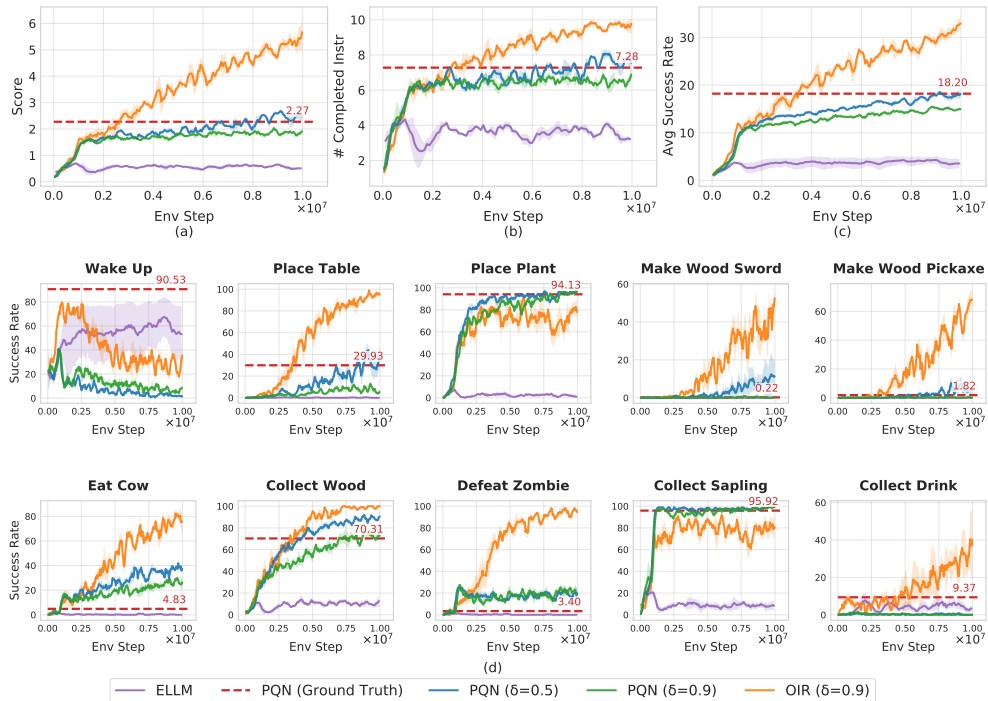

Figure 2: (a-c) Performance comparison of OIR against baseline methods measured by (a) aggregate score, (b) number of completed instructions, and (c) average success rate across all original instructions. (d) Success rates of OIR compared to baselines for individual instructions. OIR consistently outperforms baseline methods across all evaluation metrics and nearly all individual tasks. Results are averaged over three random seeds, with shaded areas representing standard errors.

instances, allowing batched interaction and large-scale querying of the LLM for instruction generation. Throughout training, the agent does not receive reward signals from the environment. For fair and direct comparison, all evaluated baselines are implemented using the same PQN algorithm backbone.

**Evaluation Protocol.** For the evaluation (conducted periodically during training), agent policies are assessed on a pre-defined suite of instructions covering three types:

1. **Original Instructions**: Standard Craftax-achievements defined by the environment (e.g., `collect wood`).

2. **Simple Variant Instructions**: Three linguistic variations per original instruction, assessing agent robustness to superficial textual differences (e.g., `collect wood` → `pick up logs from the ground.`).

3. **Complex Variant Instructions**: Three semantically enriched and compositional variants testing deeper instruction-following capabilities (e.g., `Your inventory requires wood; chop down several trees.`).

The evaluation provides external ground-truth success signals given by the environment. These signals are used *only at evaluation* to measure true instruction-following capabilities.

**Evaluation Metrics.** To quantitatively evaluate the learned instruction-following behavior, we employ three primary metrics:

- **Mean Success Rate**: Arithmetic average of the success rates across all individual instructions.

- **Mean Number of Completed Instructions**: The count of distinct instructions for which the policy achieves a success rate greater than zero.

- **Aggregate Score**: Calculated as $\exp\left(\frac{1}{N}\sum_{i=1}^{N}\log(1+s_i)\right)-1$, where $s_i$ denotes the success rate for instruction $i$. This scoring approach emphasizes improvements on instructions with lower success rates, rather than on those for which performance is already proficient.

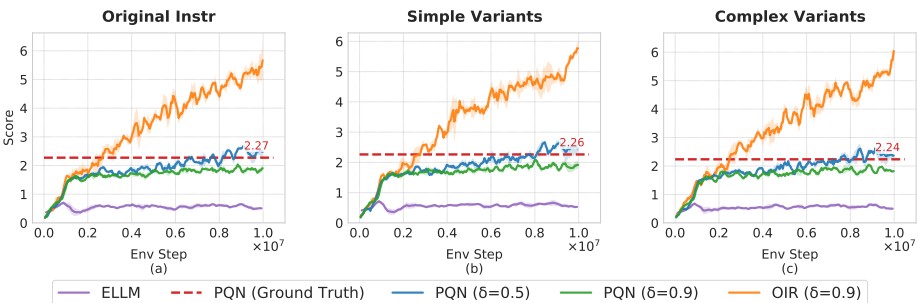

Figure 3: Generalization performance of OIR compared to baseline methods measured by aggregated score evaluation on (a) original instructions, (b) simple variant instructions, and (c) complex variant instructions. OIR demonstrates superior generalization capabilities, significantly outperforming baseline methods across all three variants. Results are averaged across three random seeds, with shaded areas denoting standard errors.

**Baselines.** We compare our proposed method against three representative baseline approaches. First, we adopt **PQN w/ cosine similarity reward**, which utilizes the PQN trained directly on the predefined *Craftax* instructions, leveraging the same cosine similarity reward function as ours. Second, we consider **PQN w/ Ground-Truth Reward**, where PQN is trained on the ground-truth sparse binary achievement rewards provided by the environment (+1 upon the achievement of the instructed task, 0 otherwise); this baseline serves as an approximate performance upper bound, as it leverages the fully accurate reward. Finally, we include **ELLM (Exploring with LLMs)** (Du et al., 2023), where exploratory goals are generated via a LLM. For fairness, we adapt ELLM to the same PQN backbone as our method and remove any domain-specific heuristics and engineering.

## 5.1 RQ1 (EFFICIENCY)

From Figures 2(a)-(c), we observe that our method OIR surpasses all baselines across every aggregated evaluation metric. Similarly, OIR demonstrates superior performance in terms of the number of completed instructions, achieving approximately 10 completed tasks compared to fewer than four tasks for other baselines. This indicates that our method not only achieves better task-specific performance but also scales better across the diverse set of instructions. The overall average success rate across instructions (33.10%) further confirms that OIR leads to consistently improved policy performance and learning efficiency compared to traditional PQN methods and ELLM.

Figure 2(d) further elucidates performance differences at the individual-instruction level. OIR learns challenging instructions, such as "Defeat Zombie" (87.78%), "Place Table" (95.83%), and "Make Wood Pickaxe" (76.59%), which other baselines completely fail to master. Such hard instructions require multi-step exploration and purposeful behavior, clearly demonstrating that the semantic guidance from open-ended instructions generated by the LLM substantially improves the exploration and learning outcomes in sparse-reward scenarios.

However, it is important to recognize that OIR does not universally outperform other methods on every individual instruction. For instance, the instruction "Wake Up" consistently yields better performance with the ground-truth PQN baseline (90.53%) compared to OIR. This reflects a trade-off inherent to our method's design: due to the fixed buffer capacity of relabeled instructions, the agent may gradually reduce proficiency on instructions that occur infrequently or lack semantic relevance to a diverse set of other instructions. Such instructions are less likely to be frequently sampled during policy updates and can decrease in performance over training. This trade-off, nonetheless, is balanced by the overall substantial improvement across a broader and richer set of instructions.

## 5.2 RQ2 (GENERALIZATION)

According to Figure 3, our OIR method consistently outperforms baselines across all instruction categories, including previously unseen variations. In contrast, baseline methods show nearly identical performance across original and previously unseen instruction variants, but at notably lower aggregate scores. Combined with the observation in Figure 2(d) that these baselines learn to successfully

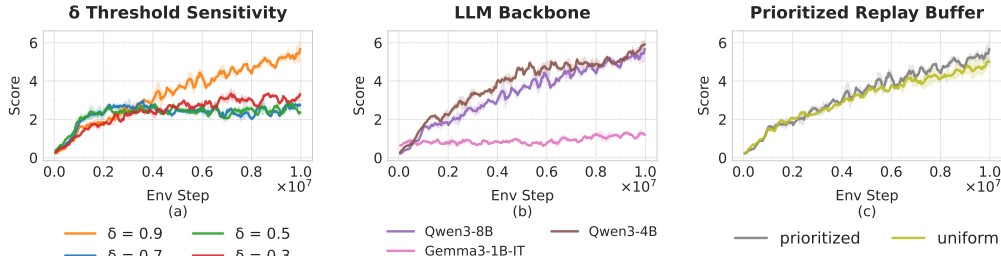

Figure 4: (a) Sensitivity to the cosine-similarity threshold $\delta$ used in the reward; (b) effect of the LLM backbone (Qwen3-8B, Qwen3-4B, Gemma3-1b-it); (c) effect of the instruction-buffer sampling strategy (prioritized instruction replay vs. uniform sampling).

complete only a small and fixed subset of instructions, we conclude that their apparently stable performance reflects an inability to learn nuanced instruction-following behaviors. Instead, they execute shallow and repetitive actions that are independent of the instruction context. Overall, these results highlight the clear advantage and scalability of OIR in effectively generalizing instruction-following policies to handle diverse and previously unseen natural-language instruction variants.

## 5.3 RQ3 (DIVERSITY)

To address RQ3, we visualize instruction embeddings with a two-dimensional t-SNE in Figure 5. Each data point corresponds to an individual instruction generated during policy training. We observe that OIR-generated instructions populate a significantly larger subregion of the embedding space, extending notably beyond the original Craftax achievements. In contrast, instructions from ELLM predominantly cluster near predefined achievements, suggesting restricted semantic variation and limited exploratory signals.

The greater semantic coverage of OIR is attributable to our hindsight instruction relabeling strategy. By generating instructions in hindsight from collected trajectories (including unsuccessful ones), OIR naturally provides a more diverse instruction distribution that semantically explains agent behaviors.

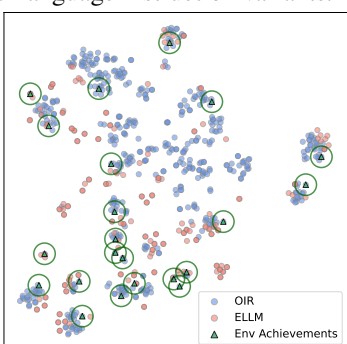

Figure 5: t-SNE visualization of semantic diversity of instructions generated by OIR compared to ELLM and environment achievements.

**Ablations.** Figure 4 presents three ablations: (a) *Cosine-similarity threshold $\delta$.* Lower $\delta$ yields denser early rewards and faster initial gains but risks over-rewarding partial progress; higher $\delta$ trades slower starts for cleaner semantics. (b) *LLM backbone for relabeling.* Qwen3-4B matches—and sometimes surpasses—Qwen3-8B, whereas a much smaller model (Gemma3-1B-IT) performs substantially worse. This pattern suggests a minimum capability threshold beyond which returns diminish; the location of this threshold likely shifts with environment difficulty. (c) *Sampling strategy.* Prioritized instruction replay eventually overtakes uniform sampling and finishes higher, consistent with the hypothesis that prioritization keeps the curriculum near the agent's learning frontier.

## 6 CONCLUSION

We proposed OIR to learn instruction-following policies by leveraging open-ended instruction relabeling with large language models. Our method automatically generates instructions from collected trajectories, effectively reducing reliance on human annotation. Experiments on the Craftax environment demonstrated improved sample efficiency, the capability to master challenging instructions, and better semantic coverage over the instruction space compared to baseline methods.

However, our method depends on the quality of instructions generated by pretrained LLMs, potentially inheriting their biases or inaccuracies. Additionally, the limited capacity of our instruction buffer can lead to the forgetting of infrequently sampled tasks. Future work should explore improved buffer management, instruction filtering, and human-in-the-loop verification to enhance robustness and applicability.

ETHICS STATEMENT

Our study trains agents entirely in the synthetic Craftax simulator and does not use human subjects, personal data, or copyrighted third-party content. Because the method relies on a pretrained LLM to generate open-ended instructions, we acknowledge potential risks from model biases or inaccurate generations. These risks are mitigated since we apply a semantic filtering step using cosine similarity reward and instruction buffer to reduce low-quality relabels. The work is intended for research on instruction-following in bounded simulators.

REPRODUCIBILITY STATEMENT

We document the environment details (including observation space, action space, reward function), and evaluation protocols in detail in Appendix B and C. We also report the hyperparameters in Table 1, including those for our algorithm and for the baselines. Finally, we provide an anonymous repository for reproducibility (link in Appendix C). We hope these materials would facilitate reproduction of our results.

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

## A  LLM USAGE

LLMs play a central role in our pipeline. We condition pretrained, open-weight models (Qwen3-4B and Qwen3-8B) on environment-generated trajectories to synthesize diverse, open-ended candidate instructions for relabeling. The exact prompts are included in our codebase and in Appendix C.5. We also use an LLM for editorial support, i.e., grammar checking and LaTeX assistance.

## B  ENVIRONMENT DETAILS

We adopt the codebase for Craftax (Matthews et al., 2024) available at `https://github.com/MichaelTMatthews/Craftax`.

### B.1  STATE SPACE

- **World grid**: discrete $H \times W$ array of block IDs.
- **Mobs**: position, health, cooldown and alive-mask for up to a fixed budget of *zombies*, *cows*, *skeletons*, *arrows*, and *growing plants*.
- **Player avatar**: position $(x, y)$, facing direction, health, food, drink, energy, fatigue, thirst, sleep flag.
- **Inventory**: integer counts (0-9) for {*wood, stone, coal, iron, diamond, sapling, wood/stone/iron pickaxe, wood/stone/iron sword*}.
- **Achievements**: per-task Boolean flags (defeat, craft, collect, etc.).
- **Time/light**: global timestep and continuous light level in $[0, 1]$.

### B.2  OBSERVATION SPACE

Agent receives a fixed-length vector obtained from a Chebyshev radius $R{=}3$ ($7{\times}7$ window) centered on the player. Each tile is one-hot-encoded and concatenated with:

- counts of each mob type at distances $1, \ldots, R$,
- the ID of the block (or mob) directly in front of the player,
- the full inventory vector,
- the four vital statistics (health, food, drink, energy).

All features are normalised to $[0, 1]$ and packed into a `Box` in $\mathbb{R}^d$.

### B.3  TEXTUAL OBSERVATION

Each step also supplies a single natural-language sentence constructed as ``Facing: ...; Nearby: ...; Inventory: ...; Status: ...'' where

1. **Facing** names the object or mob on the tile ahead,
2. **Nearby** lists non-trivial blocks/mobs in the $7{\times}7$ window, grouped by distance and alphabetised within each group,
3. **Inventory** enumerates items with non-zero count,
4. **Status** reports health, food, drink, energy and whether the avatar is sleeping.

The string is tokenised, padded to a fixed length, and embedded once per step.

### B.4  ACTION SPACE

Discrete set of $|\mathcal{A}|{=}17$ actions:

| 0 | NOOP |
|---|---|
| 1-4 | LEFT, RIGHT, UP, DOWN |
| 5 | DO / INTERACT |
| 6 | SLEEP |
| 7-10 | PLACE_STONE, PLACE_TABLE, PLACE_FURNACE, PLACE_PLANT |
| 11-13 | MAKE_WOOD_PICKAXE, MAKE_STONE_PICKAXE, MAKE_IRON_PICKAXE |
| 14-16 | MAKE_WOOD_SWORD, MAKE_STONE_SWORD, MAKE_IRON_SWORD |

These cover movement, interaction, resting, block placement, and crafting. To make the text Description more informative, we replace the generic DO / INTERACT with the actual object in front of the agent, e.g., grass.

### B.5 TESTING PROTOCOL DETAILS

**Original Instructions**  We adopt the following 22 original environment-provided instructions as the ground truth:

- – collect wood
- – place table
- – eat cow
- – collect sapling
- – collect drink
- – make wooden pickaxe
- – make wooden sword
- – place plant
- – defeat zombie
- – collect stone
- – place stone
- – eat plant
- – defeat skeleton
- – make stone pickaxe
- – make stone sword
- – wake up
- – place furnace
- – collect coal
- – collect iron
- – collect diamond
- – make iron pickaxe
- – make iron sword

**Simple & Complex Variants**  Below are examples illustrating simple and complex variants for 5 instructions:

- **Place Table**
  - *Simple:*
    - ○ A construction bench is needed; deploy one.
    - ○ The floor is a good spot for a new making-area.
    - ○ Time to set up the workbench.
  - *Complex:*
    - ○ Find a solid spot and deploy the construction apparatus there.
    - ○ Ready the zone for making things by placing the special surface.
    - ○ The workspace needs a crafting implement; position it.

- **Eat Cow**
  - *Simple:*
    - ○ Restore your energy by consuming cooked animal flesh.
    - ○ Meat from the beast is on the menu.

○ Prepared creature meat should be eaten.
  – *Complex:*
    ○ To stave off hunger, prepare and then consume the animal parts.
    ○ Sustenance can be gained from the cooked beast meat.
    ○ Replenish your energy by ingesting the prepared animal tissue.

- **Collect Sapling**
  – *Simple:*
    ○ Young trees are available; gather one.
    ○ A tree sprout is on the ground, pick it up.
    ○ Procure a small tree for future planting.
  – *Complex:*
    ○ For replanting efforts, obtain some juvenile trees.
    ○ Tree seedlings must be collected and then stored.
    ○ Environmental small tree starts are yours for the taking.

- **Collect Drink**
  – *Simple:*
    ○ Clear liquid is present; get some.
    ○ A beverage is available for pickup.
    ○ $H_2O$ for later consumption should be gathered.
  – *Complex:*
    ○ Ensure you have drinking liquid by filling a container.
    ○ Potable water needs to be collected and bottled.
    ○ From a nearby source, gather fluid for later use.

- **Make Wood Pickaxe**
  – *Simple:*
    ○ A digging tool from tree material is the goal.
    ○ Create a mining implement using wood.
    ○ Lumber can be fashioned into a digging utensil.
  – *Complex:*
    ○ The crafting table is where a digging implement of wood is made.
    ○ Combine tree-based planks and sticks for a new mining tool.
    ○ Construct a digging utensil; lumber components are required.

## C  IMPLEMENTATION DETAILS

Code for implementation of our method is available at an anonymous GitHub repository: `https://anonymous.4open.science/r/ICLR26-OIR/`.

### C.1  ALGORITHM PSEUDOCODE

We present the detailed version of the pseudocode as in Algorithm 1.

### C.2  HYPERPARAMETERS

Here we present in Table 1 the hyperparameter configurations used in all of our experiments for the baselines (PQN and ELLM) and our algorithm OIR.

### C.3  TRAINING RESOURCES & TRAINING TIME

Our experiments were conducted on a server running Ubuntu 24.04. The hardware configuration included Dual AMD EPYC 7453 28-core processors, providing a total of 112 threads. For accelerated computing, we utilized two NVIDIA A6000 GPUs and two NVIDIA A6000 Ada GPUs. The total

---

**Algorithm 2 OIR**: LLM-Guided Hindsight Instruction Relabeling (detailed version)

**Inputs**

**Require:** pretrained LLM $\mathcal{L}$; state encoder $f_{\text{state}}$; instruction encoder $f_{\text{instr}}$
**Require:** off-policy RL algorithm $\texttt{alg}$ (parameters $\theta, \psi$)
**Require:** # LLM candidates $K$; instruction buffer capacity $B_{\max}$; # parallel envs $E$
1: Initialise instruction buffer $\mathcal{B} \leftarrow \emptyset$; launch $\{\texttt{Env}_e\}_{e=1}^{E}$
2: **for** iteration $= 1, 2, \ldots$ **do**

**(1) Trajectory Collection & LLM Generation**

3:    $\mathcal{D} \leftarrow \emptyset$
4:    **for all** envs $e = 1, \ldots, E$ **in parallel do**
5:        Roll out trajectory $\tau_e = \{(s_t^e, a_t^e)\}_{t=0}^{T_e}$ with current policy $\pi_\theta$
6:        $\texttt{prompt}_e \leftarrow h(\tau_e)$                               ▷ build LLM prompt
7:        **for** $k = 1$ **to** $K$ **do**
8:            $i'_{e,k} \sim \mathcal{L}(\texttt{prompt}_e)$              ▷ generate candidate instruction
9:            $r_{e,k}^{\text{cand}} \leftarrow R(i'_{e,k}, \tau_e)$             ▷ cosine-similarity reward, Eq. 7
10:          $\mathcal{D} \leftarrow \mathcal{D} \cup \{(\tau_e, i'_{e,k}, r_{e,k}^{\text{cand}})\}$

**(2) Instruction-Buffer Maintenance**

11:    **Update** $\mathcal{B}$ with all relabeled instructions $\{\, i'_{e,k} \mid e \in [E], k \in [K]\}$ using Section 4.3
12:    **if** $|\mathcal{B}| > B_{\max}$ **then**
13:        EVICTOLDEST($\mathcal{B}, |\mathcal{B}| - B_{\max}$)

**(3) Policy Update**

14:    Update $(\theta, \psi)$ with one step of $\texttt{alg}$ on batch $\mathcal{D}$

**(4) Environment Reset / Curriculum**

15:    **for all** envs $e$ that have terminated **do**
16:        Sample $i \sim \text{UNIFORM}(\mathcal{B})$
17:        Reset $\texttt{Env}_e$ with instruction $i$

---

Table 1: Hyperparameters used in *Craftax-Classic*.

| Hyperparameter | PQN (all versions) | ELLM | OIR (ours) |
|---|---|---|---|
| Total Timesteps | $1 \times 10^7$ | $1 \times 10^7$ | $1 \times 10^7$ |
| Total Timesteps (decay) | $1 \times 10^7$ | $1 \times 10^7$ | $1 \times 10^7$ |
| Number of Environments ($N_{\text{env}}$) | 64 | 1024 | 64 |
| Steps per Environment ($N_{\text{steps}}$) | 128 | 8 | 128 |
| $\epsilon_{\text{start}}$ | 1.0 | 1.0 | 1.0 |
| $\epsilon_{\text{finish}}$ | 0.1 | 0.1 | 0.1 |
| $\epsilon$ Decay Ratio | 0.1 | 0.1 | 0.1 |
| Number of Minibatches | 4 | 4 | 4 |
| Number of Epochs | 8 | 8 | 8 |
| Input Normalization | True | True | True |
| Normalization Type | Layer Norm | Layer Norm | Layer Norm |
| Hidden Size | 512 | 512 | 512 |
| Number of Layers | 1 | 1 | 1 |
| Number of RNN Layers | 1 | 1 | 1 |
| Add Last Action | False | False | False |
| Learning Rate | $1 \times 10^{-5}$ | $1 \times 10^{-5}$ | $1 \times 10^{-5}$ |
| Max Gradient Norm | 0.5 | 0.5 | 0.5 |
| Linear LR Decay | True | True | True |
| Discount Factor ($\gamma$) | 0.99 | 0.99 | 0.99 |
| GAE $\lambda$ | 0.5 | 0.5 | 0.5 |
| Instruction Buffer Size | - | - | 10 |
| Shaping Threshold | 0.5&0.9 | 0.9 | 0.9 |
| LLM | - | Qwen3-8B | Qwen3-8B |

computational resources comprised 112 CPU threads and 512GB of system memory. Each training run for OIR and ELLM required approximately 6-8 hours to complete.

## C.4 Baseline Implementation Details

We adopt the official PQN implementation available at `https://github.com/mttga/purejaxql`. For ELLM, we closely follow the methodology presented in the original paper (Du et al., 2023), carefully implementing ELLM on top of PQN as described in the official repository (`https://github.com/yuqingd/ellm`).

To ensure a fair comparison, we remove environment-specific customizations originally included in their implementation, such as task decomposition and action-space-related checks for goal completion. Additionally, we increase the number of parallel environments to 1024, enabling batched calls to the LLM at each step. Without batching, the turnaround time would be prohibitively high.

## C.5 Prompts

---

**OIR Prompt**                                                                    `prompt:CraftAX`

**Description in Crafter:**
You are an expert Hindsight Instruction Relabeler for Minecraft agents.
You are a strict instruction generator. Under no circumstances may your output contain any movement or exploration instructions. Do **NOT** use the words `"move"`, `"explore"`, `"navigate"`, `"go to"`, or any synonyms indicating changing position.
**Environment:**

- **Resources:** wood (trees), stone, coal, iron, water, sapling (from grass)

- **Tools:** crafting table, furnace, pickaxe (wood, stone, iron), sword (wood, stone, iron)

- **Mobs:** cow, zombie, skeleton

**Principles** (max 2 each; mention an entity in every instruction):
**Mid-Level** (atomic, 1–2 steps)

- e.g., `"collect wood from the tree"`, `"collect stone"`, `"make a stone pickaxe"`, `"place crafting table"`, `"place furnace"`, `"attack zombie"`, `"attack skeleton"`, `"drink the water"`, `"wake up"`

**High-Level** (multi-step, purposeful)

- e.g., `"Make tools for collecting the iron"`, `"Collect wood then place a table, finally make a wood pickaxe"`

**Task:**
Given only a trajectory segment, output: `"Analysis": "..."`, `"Completed Instructions": "Mid-Level": ["...", "...", "...", "..."], "High-Level": ["...", "...", "...", "..."]`
**Example:**
Segment:
```
0-4:  move to tree
5:  chop tree
7-8:  place crafting table
9-10:  make wood pickaxe
11-20:  sleep
20-22:  wake up
```
Your answer: `"Analysis": "Chopped tree first, then set up crafting table, finally make wood pickaxe.", "Completed Instructions": "Mid-Level": [ "collect wood from tree", "place crafting table", "make a wood pickaxe", "sleep and wake up" ], "High-Level": [ "Prepare to collect stone", "collect tools to mine stone and coal", "prepare all tools to collect stone, then make stone pickaxe" ]`

**{{trajectory}}**

---

