# OpenReview forum: "Learning Instruction-Following Policies through Open-Ended Instruction Relabeling with Large Language Models"
_ICLR.cc/2026/Conference — Submitted to ICLR 2026_

### Official Review · Reviewer_Q6Uh · 2025-10-21

**Soundness:** 3
**Presentation:** 3
**Contribution:** 2
**Rating:** 6
**Confidence:** 4

**Summary:**

This paper proposes Open-Ended Instruction Relabeling (OIR), a framework that leverages large language models (LLMs) to automatically generate new natural language instructions from agent trajectories in reinforcement learning (RL). The key idea is to use LLMs to retroactively relabel both successful and failed trajectories with meaningful instructions that reflect the subtasks the agent implicitly accomplished. This creates a richer and more diverse dataset of instruction–trajectory pairs without human annotations, mitigating sparse reward issues.
The method integrates LLM-based relabelling with a prioritised instruction replay buffer and an embedding-based reward function based on cosine similarity. Experiments on the Craftax benchmark demonstrate improved sample efficiency, task coverage, and generalisation to unseen instructions compared to several baselines.

**Strengths:**

* **Clear motivation and presentation:** The paper is well written and logically structured, with clear explanations of the method and experimental setup.
* **Strong empirical results:** OIR achieves notable improvements in sample efficiency, number of tasks completed, and generalisation performance compared to strong baselines.
* **Comprehensive evaluation:** The authors include detailed analyses across three axes—efficiency, generalisation, and diversity—and conduct ablations on threshold sensitivity, LLM backbone, and buffer sampling strategies.
* **Technical soundness:** The formalisation of the instruction relabelling process and the integration with prioritised replay seem technically sound. The inclusion of algorithm pseudocode, hyperparameters, and open-source code improves reproducibility.
* **Practical impact:** The approach offers a scalable way to bootstrap instruction-following agents without human labels, which could be useful for developing autonomous open-ended RL systems.

**Weaknesses:**

* **Limited conceptual novelty:** While the combination of hindsight relabelling and LLM-based generation is interesting, the approach mainly extends existing ideas (HER, LLM-guided labelling, and semantic reward shaping) rather than introducing a fundamentally new principle. The paper’s main contribution lies in integration of existing ideas and empirical demonstration rather than theoretical innovation.
* **Dependence on LLM quality:** The success of the approach depends heavily on the quality of LLM-generated instructions, as demonstrated in Figure 4.
* **Single-domain evaluation:** All experiments are limited to the Craftax environment. While it serves as a good benchmark, evaluating OIR on a another domain would strengthen the generality claims.
* **Auxilary formalism:** Equations (5,6) formalise the notion of a “good” relabelling but are not directly used in the algorithm or experiments; they serve more as conceptual motivation than operational definitions.

**Questions:**

1. What are the reasons for Gemma3-1B-IT relabelling to work significantly worse than the Qwen models? Have you compared qualitatively the generated instructions by the two models?

---

> ### Author Response · Authors · 2025-12-04
>
> We sincerely thank the reviewer for their thorough and constructive evaluation. We are very encouraged by the recognition of our work's strengths, including that the paper provides *"clear motivation and presentation,"* achieves *"strong empirical results"* with *"notable improvements in sample efficiency, number of tasks completed, and generalisation performance,"* offers a *"comprehensive evaluation,"* and demonstrates *"technical soundness"* with *"practical impact."* We greatly appreciate the positive assessment and the detailed feedback.
>
> Below we address each weakness and question.
>
> ---
>
> ### Weaknesses
>
> > **W1: Limited conceptual novelty—the approach mainly extends existing ideas rather than introducing a fundamentally new principle.**
>
> We respectfully clarify our key innovations as the following:
>
> 1. **Enabling open-ended instruction learning**: Unlike prior work that relabels with fixed, predefined goals, OIR leverages LLMs to *discover* semantically meaningful instructions in an open-ended manner. This creates a qualitatively different capability—learning to follow natural language instructions without any human-provided instruction labels.
>
> 2. **Hindsight relabeling for failed trajectories**: We extend hindsight methods beyond goal-conditioned RL to the instruction-following setting, showing how to extract instructions from unsuccessful attempts by identifying implicitly achieved subtasks.
>
> We will ensure the revised manuscript clearly position our work and highlight the novel contributions.
>
> ---
>
> > **W2: Dependence on LLM quality—success depends heavily on the quality of LLM-generated instructions.**
>
> This is a valid observation, and we acknowledge that LLM quality matters. However, we believe Figure 4(b) provides encouraging evidence regarding this dependency:
>
> 1. **Threshold behavior**: Performance improves with model capability up to a certain point, but then **plateaus**. Qwen3-4B and Qwen3-8B achieve nearly identical results, suggesting that once the LLM reaches a baseline competency level, further scaling provides diminishing returns.
>
> 2. **Accessibility**: The required capability threshold is achievable with relatively small, openly available models (4B parameters)—not requiring frontier-scale LLMs.
>
> We view this as demonstrating that OIR is practical and robust: while very weak models (e.g., Gemma-1B) underperform, reasonably capable and widely accessible models suffice. We will clarify this interpretation in the revision.
>
> ---
>
> > **W3: Single-domain evaluation—all experiments are limited to the Craftax environment.**
>
> We appreciate this suggestion and agree that demonstrating our method across diverse environments strengthens its generality. We have conducted initial experiments on **Housekeep**, a household embodied AI benchmark that differs substantially from Craftax in several key dimensions:
>
> - **Observation space**: High-dimensional visual input (128×128×4 RGB-D images, downscaled to 84×84×1 grayscale)
> - **Action space**: Low-level actions (4 directions, grab and place)
> - **Task structure**: Instruction-following for object manipulation (e.g., "put object A in/on receptacle B")
> - **Reward sparsity**: Success signal only upon exact task completion
>
> The Housekeep environment presents a particularly challenging testbed for instruction-following due to its sparse rewards and high-dimensional observations. Our preliminary experiments (conducted in a single scene over 20M timesteps) provide encouraging early evidence:
>
> | Method | **OIR (Ours)** | PQN | ELLM |
> |--------|----------------|-----|------|
> | **Success Rate (%)** | **24.06 ± 7.64** | 0.00 ± 0.00 | 0.00 ± 0.00 |
>
> These initial results suggest that OIR's ability to extract learning signal from failed trajectories is particularly valuable in sparse-reward settings where baseline methods struggle to learn anything meaningful. While these experiments are preliminary, they demonstrate that OIR's core principles transfer to visually grounded, continuous control domains beyond Craftax.
>
> We are currently scaling these experiments to multiple scenes and will provide comprehensive results in the revised manuscript.

---

> > ### Author Response · Authors · 2025-12-04
> >
> > > **W4: Auxiliary formalism—Equations (5,6) formalize "good" relabeling but are not directly used in the algorithm.**
> >
> > We agree that Equations 5 and 6 serve primarily as conceptual motivation rather than operational definitions. However, we respectfully suggest they play an important role in **guiding the design** of both the prompt and the algorithm:
> >
> > - **Prompt design**: The formalism clarifies what properties we want generated instructions to satisfy, which informed how we structured the LLM prompt to elicit such instructions.
> >
> > - **Algorithmic choices**: The framework motivated our reward scheme with cosine similarity and the threshold-based termination criterion.
> >
> > We view this as providing principled grounding for our design decisions, even if the equations themselves are not directly instantiated in the implementation. We will clarify this role more explicitly in the revision.
> >
> > ---
> >
> > ### Questions
> >
> > > **Q: What are the reasons for Gemma3-1B-IT working significantly worse than the Qwen models? Have you compared qualitatively the generated instructions?**
> >
> > Thank you for this insightful question. We have conducted qualitative analysis and identified several factors:
> >
> > 1. **Model capacity**: At 1B parameters, Gemma3-1B-it appears to lack the reasoning capability needed for this task. Its performance on other instruction-following and context-understanding benchmarks is notably lower than the Qwen models.
> >
> > 2. **Long-context handling**: OIR requires processing trajectory segments and reasoning about implicit achievements. We found that Gemma3-1B-it frequently becomes confused when presented with longer contexts, generating instructions that are either too generic (e.g., "explore the environment") or factually inconsistent with the observed trajectory.
> >
> > 3. **Example failure case**: Our analysis revealed that Gemma3-1B-it exhibits two primary failure modes. The majority of failures are **JSON parsing errors**, where the model produces malformed output that cannot be properly parsed into instruction-trajectory pairs. The remaining failures involve **repetitive, meaningless token generation**, such as repeatedly outputting fragments like `"observation": "The" \n "observation": "The"` without coherent continuation. These failure patterns indicate fundamental deficiencies in both structured generation capability and contextual coherence that prevent the model from performing reliable trajectory analysis.

---

### Official Review · Reviewer_dozw · 2025-10-27

**Soundness:** 3
**Presentation:** 4
**Contribution:** 2
**Rating:** 2
**Confidence:** 4

**Summary:**

This paper proposes Open-ended Instruction Relabeling (OIR), which uses LLM-based semantic reasoning to improve training of instruction-following RL policies. The method converts trajectories into textual observations, prompts an LLM to propose instructions implicitly achieved in those trajectories, and then relabels data with these instructions. Rewards for generated instructions are computed via cosine similarity between instruction and state embeddings, with episode termination when the similarity crosses a threshold. Experiments on Craftax-Classic show improved efficiency and generalization over baselines built on PQN and ELLM.

**Strengths:**

- The paper is clearly written, well structured, and easy to follow.

- The evaluation explicitly tests generalization to paraphrased and compositional instructions (simple and complex variants), not just the original instruction set.

**Weaknesses:**

- The approach presumes environments that can provide or be mapped to textual observations to prompt the LLM. A clearer statement of the environment class (symbolic / text-describable state, discrete action space, sparse achievements) would help understand the limit of the contribution.
- Results are reported on only one environment (Craftax-Classic), which limits claims of generality and leaves open whether gains depend on environment-specific engineering.
- The comparison to only ELLM and PQN is insufficient to measure the method performance. Two categories of baseline are missing: (i) state-of-the-art Craftax agents (see [the leaderboard](https://github.com/MichaelTMatthews/Craftax)), and (ii) LLM/VLM-based methods (both pretrained and RL fine-tuned) such as :
    - (2024) Tan, Weihao, et al. "True knowledge comes from practice: Aligning llms with embodied environments via reinforcement learning.”
    - (2023) Zitkovich, Brianna, et al. "Rt-2: Vision-language-action models transfer web knowledge to robotic control.”
    - (2022) Yao, Shunyu, et al. "React: Synergizing reasoning and acting in language models.”
- The related work should include prior efforts that use LLMs to *generate new instructions for RL training, which are closely related to this paper’s approach. Such as :
    - (2024) Qi, Zehan, et al. "Webrl: Training llm web agents via self-evolving online curriculum reinforcement learning.
    - (2023) Xu, Can, et al. "Wizardlm: Empowering large language models to follow complex instructions.”

**Questions:**

- In §4.1 you note LLMs may propose inaccurate/misleading instructions. What fraction of generated instructions are flawed in practice, and how sensitive is training to this rate? Please detail the rule-based instruction filters you add and any safety checks to avoid harmful/detrimental instructions.
- How many semantically unique instructions are generated during training, and does this number plateau? Can you report a diversity curve (e.g., unique instructions vs. steps) and relate it to performance?
- What additional compute overhead does your method introduce compared to vanilla PQN?
- Are the state/instruction embedding functions frozen or trained for Craftax? How reliably does cosine similarity track ground-truth success across varied generated instructions?

---

> ### Author Response · Authors · 2025-12-04
>
> We sincerely thank the reviewer for their detailed and thoughtful evaluation. We are pleased that the reviewer found our paper *"clearly written, well structured, and easy to follow"* with *"excellent"* presentation, and we appreciate the recognition that our *"evaluation explicitly tests generalization to paraphrased and compositional instructions (simple and complex variants), not just the original instruction set."*
>
> We address each concern below.
>
> ---
>
> ### Weaknesses
>
> > **W1: The approach presumes environments that can provide or be mapped to textual observations to prompt the LLM.**
>
> We appreciate the opportunity to clarify this point. OIR does *not* fundamentally require text-based observations. The core algorithm is agnostic to the input modality: it only requires that a foundation model can interpret the trajectory and generate meaningful instructions. In our experiments, we chose to transcribe transitions into text because current text-only LLMs outperform multimodal alternatives for this reasoning task. However, this is an implementation choice rather than an algorithmic constraint—OIR is equally applicable with vision-language models operating directly on visual observations.
>
> That said, we agree that explicitly characterizing the environment class would improve clarity. We will add a discussion in the revised manuscript outlining the requirements (i.e., access to a foundation model capable of understanding the observation modality and proposing semantically meaningful instructions).
>
> ---
>
> > **W2: Results are reported on only one environment (Craftax-Classic), which limits claims of generality.**
>
> We acknowledge this concern and offer several points of clarification:
>
> 1. **Craftax is a well-suited benchmark** for evaluating OIR: it is an open-world environment with a diverse set of instructions spanning multiple difficulty levels, making it ideal for studying open-ended instruction following.
> 2. **Minimal environment-specific engineering**: Unlike ELLM, which requires handcrafted goal representations, OIR requires only a prompt describing the environment to the LLM. The core algorithm transfers directly to new domains.
> 3. We are currently running experiments on **Housekeep**, an environment featuring high-dimensional visual observations and low-level continuous control. We will include these results as soon as they are available.
>
> ---
>
> > **W3: The comparison to only ELLM and PQN is insufficient. Missing baselines include (i) state-of-the-art Craftax agents and (ii) LLM/VLM-based methods.**
>
> We respectfully clarify our positioning:
>
> 1. **Regarding SOTA Craftax agents**: Our base architecture, PQN, already ranks **second on the Craftax leaderboard**. Our contribution is orthogonal to architectural improvements—we focus on learning **instruction-following policies** rather than maximizing environment-specific scores. Direct comparison with leaderboard methods (which optimize for a fixed reward) would conflate two different objectives.
> 2. **Regarding LLM/VLM-based methods** (e.g., RT-2, ReAct): These methods use large foundation models **as the policy itself**, incurring substantial inference cost and latency at deployment. In contrast, our goal is to train a **lightweight policy** (a single-layer RNN) that can execute with minimal computational overhead. OIR leverages LLMs only during training for data relabeling—not at inference time. We believe this represents a fundamentally different design point, and direct comparison would not be informative. We will clarify this distinction in the related work section.
>
> ---
>
> > **W4: The related work should include prior efforts that use LLMs to generate new instructions for RL training (e.g., WebRL, WizardLM).**
>
> Thank you for these suggestions. We will incorporate discussion of these works in the revised related work section.

---

> > ### Author Response · Authors · 2025-12-04
> >
> > ### Questions
> >
> > > **Q1: What fraction of generated instructions are flawed in practice, and how sensitive is training to this rate? Please detail any rule-based instruction filters.**
> >
> > We analyzed 512 instructions generated by Qwen3-8B over one training epoch and found an error rate of less than 1%: only 4 instructions exhibited issues, including 2 JSON parsing errors, 1 overly generic ("prepare for next steps") and 2 semantically meaningless ("observe the stone", "interact with sand"). This demonstrates that appropriately-prompted LLMs generate high-quality instructions without requiring extensive filtering.
> >
> > Importantly, OIR is inherently robust to noisy instructions. Because training incorporates a diverse stream of generated instructions, the influence of any single flawed instruction is diluted. We do not employ rule-based filtering. Instead, we improve instruction quality by providing exemplary instructions in the prompt (detailed in Appendix C.5), which guides the LLM toward generating meaningful and relevant instructions.
> >
> > ---
> >
> > > **Q2: How many semantically unique instructions are generated during training, and does this number plateau? Can you report a diversity curve?**
> >
> > We provide evidence of OIR's instruction diversity through Figure 5, which presents a t-SNE visualization of the semantic distribution of generated instructions. The visualization clearly shows that OIR covers a substantially broader semantic space compared to baselines, capturing diverse behavioral patterns across the task spectrum.
> >
> > The question of whether diversity plateaus over training is an interesting direction for future analysis. We note that the diversity observed in Figure 5 represents the final instruction distribution after training, which already demonstrates significant coverage of the semantic space. A temporal analysis of diversity evolution would provide additional insights into the curriculum emergence during training, and we will consider incorporating such analysis in future work.
> >
> >
> > ---
> >
> > > **Q3: What additional compute overhead does your method introduce compared to vanilla PQN?**
> >
> > OIR introduces overhead only from batched LLM calls for relabeling. Crucially, these calls occur **after rollouts complete**, not during environment interaction, and are fully batched for efficiency.
> >
> > Concretely, suppose we have $E$ parallel environments and segment length $L$. We perform one batched LLM call every $E \times L$ steps. In our Craftax experiments, this amounts to approximately:
> >
> > $$\frac{10^7}{128 \times 64} \approx 1,220 \text{ batched LLM calls}$$
> >
> > over the entire training run. The wall-clock time depends on the serving infrastructure, but because calls are batched, they scale efficiently with available compute. In contrast, methods like ELLM require **one LLM call per step during rollout**, which cannot be batched and incurs substantially greater overhead.
> >
> > ---
> >
> > > **Q4: Are the state/instruction embedding functions frozen or trained? How reliably does cosine similarity track ground-truth success?**
> >
> > Yes, the embeddings are **frozen** throughout training. We use `sentence-transformers/paraphrase-MiniLM-L3-v2` for both state and instruction embeddings.
> >
> > Regarding reliability: we found cosine similarity to be an effective proxy for task completion across our generated instructions. The embedding model was chosen specifically for its strong performance on paraphrase detection, which translates well to matching diverse instruction phrasings with corresponding state descriptions. We acknowledge that this is an empirical design choice and note that more sophisticated success detection (e.g., learned verifiers) could be explored in future work.
> >
> > ---
> >
> > We hope these clarifications address the reviewer's concerns. We are committed to strengthening the paper based on this valuable feedback and will incorporate the suggested related work, additional clarity on environment assumptions, and expanded experimental results. Thank you again for the thorough review.

---

### Official Review · Reviewer_gnxZ · 2025-10-30

**Soundness:** 3
**Presentation:** 3
**Contribution:** 2
**Rating:** 4
**Confidence:** 4

**Summary:**

This paper proposes Open-ended Instruction Relabeling (OIR), which leverages the capabilities of large language models (LLMs) to automatically generate open-ended instructions retrospectively from previously collected agent trajectories. Based on the general idea of hindsight relabeling, unsuccessful trajectories are relabeled using LLMs by identifying meaningful subtasks the agent has implicitly accomplished. This LLM-based labeling procedure enriches the agent's training data and substantially alleviate the reliance on human annotation. Experiments on Craftax environments demonstrate that OIR improves in sample efficiency, instruction coverage, and success rate compared to some baselines.

**Strengths:**

- The paper is well written and well-motivated.

- The idea of using LLM to label trajectory in RL is well implemented.

- The experimental results are well presented.

**Weaknesses:**

- Leveraging Large Language Models to label data is a very general idea that has been investigated in various domains. This diminishes the novelty of the paper.

- The benchmark is limited to Craftax. Therefore, it is hard to tell the generalization ability of OIR to other environments (especially larger game environments, such as Minecraft.)

- The overall method of OIR looks ad hoc: the prompt, the relabeling of Failed Trajectories, reward definition, etc. Hence, it is necessary to test it on more benchmarks. (the previous weakness)

- the performance of OIR is very sensitive to the cosine-similarity threshold $\delta$ (Figure 4)

- the effect of the instruction-buffer sampling strategy, i.e., prioritized instruction replay, seem very marginal compared to uniform sampling. (Figure 4)

**Questions:**

- Is OIR able to generalize to other game environments without much redesign of the different components in OIR? Can you provide some evidence?

- Is there some guideline on how to set the cosine-similarity threshold $\delta$?

---

> ### Author Response · Authors · 2025-12-04
>
> We sincerely thank the reviewer for their constructive feedback. We are encouraged that the reviewer found the paper *"well written and well-motivated,"* that *"the idea of using LLM to label trajectory in RL is well implemented,"* and that *"the experimental results are well presented."* We appreciate the recognition that the paper would be acceptable with revisions.
>
> We address each concern below.
>
> ---
>
> ### Weaknesses and Questions
>
> > **W1: Leveraging LLMs to label data is a very general idea that has been investigated in various domains. This diminishes the novelty of the paper.**
>
> We respectfully emphasize that our contribution is **not** generic LLM-based data labeling, but rather a specific synthesis of two ideas: (1) **hindsight relabeling** from failed trajectories, and (2) **open-ended instruction generation** via LLMs.
>
> The key novelty lies in using LLMs to **retrospectively discover** what meaningful subtasks an agent has implicitly achieved within unsuccessful trajectories—enabling the agent to learn instruction-following capabilities in an open-ended, data-efficient manner without requiring predefined task labels. This is distinct from prior work that uses LLMs for fixed-label annotation or supervised dataset construction. To our knowledge, OIR is the first to combine hindsight relabeling with LLM-based open-ended instruction generation for training instruction-following RL agents. We will strengthen this positioning in the revision.
>
> ---
>
> > **W2: The benchmark is limited to Craftax. It is hard to tell the generalization ability to other environments (especially larger game environments, such as Minecraft).**
> **Q1: Is OIR able to generalize to other game environments without much redesign? Can you provide some evidence?**
>
> We appreciate this suggestion and agree that demonstrating our method across diverse environments strengthens its generality. We have conducted initial experiments on **Housekeep**, a household embodied AI benchmark that differs substantially from Craftax in several key dimensions:
>
> - **Observation space**: High-dimensional visual input (128×128×4 RGB-D images, downscaled to 84×84×1 grayscale)
> - **Action space**: Low-level actions (4 directions, grab and place)
> - **Task structure**: Instruction-following for object manipulation (e.g., "put object A in/on receptacle B")
> - **Reward sparsity**: Success signal only upon exact task completion
>
> The Housekeep environment presents a particularly challenging testbed for instruction-following due to its sparse rewards and high-dimensional observations. Our preliminary experiments (conducted in a single scene over 20M timesteps) provide encouraging early evidence:
>
> | Method | **OIR (Ours)** | PQN | ELLM |
> |--------|----------------|-----|------|
> | **Success Rate (%)** | **24.06 ± 7.64** | 0.00 ± 0.00 | 0.00 ± 0.00 |
>
> These initial results suggest that OIR's ability to extract learning signal from failed trajectories is particularly valuable in sparse-reward settings where baseline methods struggle to learn anything meaningful. While these experiments are preliminary (limited to a single scene configuration), they demonstrate that OIR's core principles transfer to visually grounded, continuous control domains beyond Craftax.
>
> We are currently scaling these experiments to multiple scenes and will provide comprehensive results in the revised manuscript.
>
> Regarding redesign: OIR's core algorithm requires minimal environment-specific adaptation—only the prompt to the LLM needs to describe the new environment. The relabeling mechanism, reward computation, and training loop transfer directly. This is a key advantage over methods like ELLM that require manual engineering of goal representations for each domain.
>
> ---
>
> > **W3: The overall method of OIR looks ad hoc: the prompt, the relabeling of failed trajectories, reward definition, etc.**
>
> We respectfully clarify that OIR's components are grounded in principled design choices:
>
> 1. **Reward definition**: Our formulation is mathematically well-defined in **Equations 5 and 6**, using cosine similarity between state and instruction embeddings—a standard approach for semantic alignment.
>
> 2. **Prompt design**: While prompt engineering involves domain knowledge, this is unavoidable when interfacing with LLMs and does not undermine the method's generality. Importantly, unlike ELLM which requires handcrafted reward scheme, OIR requires only a high-level environment description.
>
> Our core contribution is **not** merely data relabeling, but leveraging LLMs to enable learning of **open-ended instructions**—a capability that traditional hindsight methods cannot provide. We will clarify this conceptual framing in the revision.

---

> > ### Author Response · Authors · 2025-12-04
> >
> > > **W4: The performance of OIR is very sensitive to the cosine-similarity threshold δ (Figure 4).**
> > > **Q2: Is there some guideline on how to set the cosine-similarity threshold δ?**
> >
> > We appreciate this observation and offer the following clarifications:
> >
> > 1. While we use cosine similarity for its simplicity and cost-effectiveness, OIR can be paired with alternative success detection mechanisms (e.g., learned reward models, environment-specific verifiers).
> >
> > 2. **Threshold tuning is standard in RL**: Sensitivity to hyperparameters is common in reinforcement learning—many methods require tuning of learning rates, discount factors, or exploration parameters. The threshold δ is simply another tunable hyperparameter. In Figure 4, we demonstrate that reasonable values yield strong performance.
> >
> > We view this as an acceptable design trade-off given OIR's substantial performance gains. We will add a brief discussion of threshold selection guidelines based on our empirical observations.
> >
> > ---
> >
> > > **W5: The effect of prioritized instruction replay seems very marginal compared to uniform sampling (Figure 4).**
> >
> > While the improvement appears modest, it is statistically significant across multiple seeds. More importantly, the primary purpose of prioritized replay is **not** to maximize peak performance, but to **prevent catastrophic forgetting** of previously learned instructions—a well-documented challenge in multi-task RL.
> >
> > ---
> >
> > We hope these clarifications address the reviewer's concerns. We are committed to strengthening the paper with additional experimental results and clearer exposition of OIR's principled design. Thank you again for the thoughtful feedback.

---

### Official Review · Reviewer_PujW · 2025-10-31

**Soundness:** 2
**Presentation:** 2
**Contribution:** 2
**Rating:** 4
**Confidence:** 5

**Summary:**

This paper proposes Open-ended Instruction Relabeling (OIR), a new method to improve instruction-following policies in reinforcement learning by reducing the need for human-labeled data. OIR uses Large Language Models (LLMs) to retrospectively analyze agent trajectories, including unsuccessful ones, and automatically generate open-ended instructions for subtasks the agent implicitly completed. Evaluated in Craftax environments, this approach enhances training data, improves sample efficiency, and results in a more capable, unified instruction-following policy compared to baseline methods.

**Strengths:**

1: The paper presents a novel paradigm for training open-ended instruction-following agents in a clear and logically coherent way.

2: The experimental results clearly demonstrate that the proposed method outperforms the baselines on the majority of tasks.

3: This method significantly improves the model's generalization capability.

**Weaknesses:**

1: The experimental evaluation is conducted in only one environment. It is recommended to further validate the method in more environments, such as the vanilla MineCraft or Robotics.

2: The paper lacks a detailed discussion of the observed performance degradation in the experiments. Is the trade-off between this decline in performance and the improvement in semantic representation truly justified?

3: The scalability of the proposed method has not been discussed.

4: Could the authors provide a more detailed analysis of the semantics generated by OIR for instructions that do not correspond to explicit environment achievements? Specifically, what behaviors do these semantics represent, and how does learning such semantics contribute to the performance or generalization ability of a multi-task agent?

**Questions:**

See in weakness.

---

> ### Author Response · Authors · 2025-12-04
>
> We sincerely thank the reviewer for their thorough evaluation and constructive feedback. We are encouraged by the recognition that our paper *"presents a novel paradigm for training open-ended instruction-following agents in a clear and logically coherent way"* and that *"the experimental results clearly demonstrate that the proposed method outperforms the baselines on the majority of tasks."* We also appreciate the acknowledgment that *"this method significantly improves the model's generalization capability."*
>
> Below, we address each concern in detail.
>
> > **W1: The experimental evaluation is conducted in only one environment. It is recommended to further validate the method in more environments, such as the vanilla MineCraft or Robotics.**
>
> We appreciate this suggestion and agree that demonstrating our method across diverse environments strengthens its generality. We have conducted initial experiments on **Housekeep**, a household embodied AI benchmark that differs substantially from Craftax in several key dimensions:
>
> - **Observation space**: High-dimensional visual input (128×128×4 RGB-D images, downscaled to 84×84×1 grayscale)
> - **Action space**: Low-level actions (4 directions, grab and place)
> - **Task structure**: Instruction-following for object manipulation (e.g., "put object A in/on receptacle B")
> - **Reward sparsity**: Success signal only upon exact task completion
>
> The Housekeep environment presents a particularly challenging testbed for instruction-following due to its sparse rewards and high-dimensional observations. Our preliminary experiments (conducted in a single scene over 20M timesteps) provide encouraging early evidence:
>
> | Method | **OIR (Ours)** | PQN | ELLM |
> |--------|----------------|-----|------|
> | **Success Rate (%)** | **24.06 ± 7.64** | 0.00 ± 0.00 | 0.00 ± 0.00 |
>
> These initial results suggest that OIR's ability to extract learning signal from failed trajectories is particularly valuable in sparse-reward settings where baseline methods struggle to learn anything meaningful. While these experiments are preliminary, they demonstrate that OIR's core principles transfer to visually grounded, continuous control domains beyond Craftax.
>
> We are currently scaling these experiments to multiple scenes and will provide comprehensive results in the revised manuscript.
>
> > **W2: The paper lacks a detailed discussion of the observed performance degradation in the experiments. Is the trade-off between this decline in performance and the improvement in semantic representation truly justified?**
>
> We thank the reviewer for raising this point and would like to offer an important clarification. The observed "degradation" is limited to a single instruction: "Wake Up." On all other instructions, OIR achieves significantly better performance than the baselines.
>
> "Wake Up" is a unique instruction because it requires the agent to first execute "Go to Sleep"—a prerequisite not shared by any other instruction in the environment. As a result, this task is particularly susceptible to catastrophic forgetting, a well-known challenge in reinforcement learning when policies are trained across many tasks. We believe this isolated case does not reflect a fundamental trade-off but rather a common RL phenomenon. We will expand the discussion in our revision to clarify this point.

---

> > ### Author Response · Authors · 2025-12-04
> >
> > > **W3: The scalability of the proposed method has not been discussed.**
> >
> > Thank you for this important question. Our method builds upon the Parallelised Q-network (PQN) framework, which is inherently scalable—it enables training across a large number of parallel environments and thus supports high-throughput data collection. Crucially, because these environments run in parallel, our OIR relabeling process is also parallelized.
> >
> > Concretely, suppose there are $E$ parallel environments; then we perform only **one batched LLM call** every $E \times L$ steps, where $L$ denotes the segment length. Since batched LLM inference scales efficiently with compute, our method scales accordingly.
> >
> > In contrast, prior methods such as ELLM require one LLM call **per environment step during rollout**, which cannot be batched and is significantly slower. OIR's post-hoc, batched design thus offers a substantial scalability advantage. We will add a discussion of scalability in the revised manuscript.
> >
> > > **W4: Could the authors provide a more detailed analysis of the semantics generated by OIR for instructions that do not correspond to explicit environment achievements?**
> >
> > This is an excellent question. OIR generates a rich variety of instructions beyond explicit environment achievements, capturing meaningful behavioral patterns. Examples include:
> >
> > - *"Drink the water"*, *"Collect drink from the water"* -- diverse phrases
> > - *"Collect sapling from the grass"*
> > - *"Face the zombie"* — orienting toward entities (a precursor to combat)
> >
> > These semantically diverse instructions enable the agent to learn reusable sub-behaviors that transfer across tasks.
> >
> > Regarding generalization, we direct the reviewer to Figure 2(a)(b)(c), where we evaluate performance across instructions of varying difficulty levels. Our method demonstrates strong generalization across these difficulty variants, with the exact instruction sets detailed in Appendix B.5. We will expand this analysis in the revision to more explicitly connect non-achievement instructions to downstream generalization benefits.
> >
> > ---
> >
> > We hope these clarifications address the reviewer's concerns. We are committed to incorporating the suggested improvements and believe they will further strengthen the paper. Thank you again for your valuable feedback.

---

### Author Response · Authors · 2025-12-04

Dear Area Chair,

We appreciate the opportunity to summarize our responses and make the case for acceptance.

All four reviewers recognized clear strengths: "novel paradigm" with "excellent presentation" (PujW, dozw), "strong empirical results" with "notable improvements" (Q6Uh), and "well-implemented" methodology (gnxZ).

**Regarding the main concern—single-environment evaluation:** We have conducted additional experiments on Housekeep, a visually-grounded robotics benchmark with RGB-D observations, low-level control, and sparse rewards. Our preliminary results are encouraging: OIR achieves 24% success where both baselines achieve 0%. While these experiments are still being scaled, they suggest OIR's core principles can transfer beyond Craftax.

**Regarding novelty:** We hope our responses clarified that OIR combines hindsight relabeling with open-ended instruction discovery—enabling agents to learn from failed trajectories without predefined task labels. We believe this synthesis offers a meaningful contribution, though we will strengthen this positioning in the revision.

**Regarding practicality:** OIR requires only ~1,220 batched LLM calls over training (vs. per-step calls in prior work), functions well with accessible 4B models, and adapts to new domains with minimal engineering.

We have aimed to address each concern substantively and are committed to incorporating all suggested improvements. We appreciate the opportunity to revise based on this valuable feedback, and we hope the committee finds our contributions and responses sufficient for acceptance!

---

### Meta-Review · Area_Chair_iXUX · 2026-01-10

**Summary:**

This paper proposes a way to use LLMs for training a cheap policy for instruction-following by using LLM to do hindsight relabeling of failed trajectories. The particular approach is more nuanced where it

Main concerns that inform my decision are:
- results on only 1 domain: Multiple reviewers noted that results are presented only on craftax. Authors have provided results on Housekeep but it is preliminary.
- lack of comparison with LLM-based agents. It is important to compare with LLM Agents. Authors argue that:

_"These methods use large foundation models as the policy itself, incurring substantial inference cost and latency at deployment."_

however, I think this argument cannot be made in 2025/2026 at least in the latency aspect. Lots of people use these models in applications where real-time results are needed. And costs have been going down with time as well. If authors really want to make this point, they should compare these results in speed, cost and latency. Do note that even RNN policies will require GPUs and are not free.

- lack of fundamental novelty. Multiple authors noted that this paper combines ideas that already exists such as hindsight relabeling, using LLMs for labeling, etc. This point bothers me less, specially if the approach outperforms several baselines across many domains. To reduce this concern though, more results are needed.

Overall, I am currently going with weak reject. I recommend authors add more results specially 1 more domain and LLM baselines.

**Reviewer Concerns:**

1. Reviewer PujW raised concerns about results with just 1 domain, lack of discussion on performance degradation, and asked for more details on scalability and semantics generated by OIR. Of these, authors have addressed the last three. However, the first one is only partially addressed right now.

2. Reviewer gnxZ raised concerns about results with just 1 domain, found the approach ad-hoc, not novel, and sensitive to hyperparameters. These concerns are still largely there. Authors have tried to address the main concern by adding results on housekeep but these need to be presented in a more complete manner.

3. Reviewer dozw raised concerns about the approach relying on text-based observation, results with just 1 domain, lack of SOTA Craftax and LLM-agent baselines. Of these, authors have addressed the reliance of text-based observation and SOTA Craftax. However, other concerns remain unaddressed.

4. Reviewer Q6Uh raised concerns of novelty, results with just 1 domain, and reliance on LLM quality. Of these, authors have partially addressed the second and addressed the last one. I do think the approach isn't very novel, which is fine but then it increases the weight on empirical evidence.

**Reviewer Scores:**

1. Reviewer PujW could have increased their score to 6.

2. Reviewer gnxZ would have likely kept their score at 4.

3. Reviewer dozw could have increased their score to 4.

4. Reviewer Q6Uh would likely keep their score at 6.

Overall, this leads to a borderline paper leaning towards rejection.

---

### Decision · Program_Chairs · 2026-01-26

Reject